# Expression Profiles of lncRNAs and mRNAs in the Mouse Brain Infected with Pseudorabies Virus: A Bioinformatic Analysis

**DOI:** 10.3390/v17040580

**Published:** 2025-04-17

**Authors:** Yanwei Li, Teng Tu, Yan Luo, Xueping Yao, Zexiao Yang, Yin Wang

**Affiliations:** Key Laboratory of Animal Diseases and Human Health of Sichuan Province, College of Veterinary Medicine, Sichuan Agricultural University, Chengdu 611130, China; 2022203032@stu.sicau.edu.cn (Y.L.); 2019303092@stu.sicau.edu.cn (T.T.); 41187@sicau.edu.cn (Y.L.); 13577@sicau.edu.cn (X.Y.); 13643@sicau.edu.cn (Z.Y.)

**Keywords:** pseudorabies virus, RNA-seq, lncRNA, brain

## Abstract

Pseudorabies virus (PRV) is a pathogen that causes severe neurological dysfunction in the host, leading to extensive neuronal damage and inflammation. Despite extensive research on the neuropathogenesis of α-herpesvirus infections, many scientific questions remain unresolved, such as the largely unknown functions of long non-coding RNAs (lncRNAs) in herpesvirus-infected nervous systems. To address these questions, we used RNA sequencing (RNA-seq) to investigate the expression profiles of lncRNAs and mRNAs in the brains of mice infected with PRV. Through bioinformatic analysis, we identified 316 differentially expressed lncRNAs and 886 differentially expressed mRNAs. We predicted the biological functions of these differentially expressed lncRNAs and mRNAs using the Gene Ontology (GO) and Kyoto Encyclopedia of Genes and Genomes (KEGG) databases, and the results showed that the differentially expressed transcripts were mainly involved in the innate immune response. Finally, we validated the differential expression trends of lncRNAs and mRNAs using quantitative real-time PCR (q-PCR), which were consistent with the sequencing data. To our knowledge, this is the first report analyzing the lncRNA expression profile in the central nervous system (CNS) of mice infected with PRV. Our findings provide new insights into the roles of lncRNAs and mRNAs during PRV infection of the host CNS.

## 1. Introduction

Pseudorabies, also known as Aujeszky’s disease, is an acute infectious disease caused by pseudorabies virus (PRV). In pigs, PRV infection often leads to encephalitis or reproductive disorders, causing significant losses to the swine industry [1]. Although the use of attenuated live vaccines has eradicated PRV in many countries, recent studies have shown that PRV is widely present in wild boar populations, making them a potential source of infection for domestic pigs [2]. Additionally, PRV, like other α-herpesviruses, has the ability to establish latent infections in the nervous system. Hosts with latent infections do not release viral particles into the environment [3]. Therefore, PRV still poses a continuous potential threat, and the control of pseudorabies remains a significant challenge.

Pseudorabies virus (PRV), also known as Suid herpesvirus 1, belongs to the genus Varicellovirus within the subfamily Alphaherpesvirinae of the family Herpesviridae. PRV is a pantropic virus that can infect a variety of cell types [4]. Moreover, it shares a common feature with other α-herpesviruses: its strong neurotropism, which allows it to invade the peripheral nervous system (PNS) and central nervous system (CNS), leading to acute neurological diseases or establishing latency in neurons under specific conditions [5]. In adult pigs, PRV typically establishes asymptomatic latent infections in PNS ganglia. In contrast, PRV infection in neonatal piglets and non-natural hosts, such as ruminants and rodents, often leads to fatal encephalitis and subsequent host death [6]. Furthermore, recent studies have shown that PRV can cause encephalitis in humans. Researchers identified PRV-specific sequences in the tissues of patients with acute encephalitis using metagenomic sequencing and isolated the first human PRV strain (hSD-1/2019) from cerebrospinal fluid [7,8,9]. Despite extensive research on neurotropic viruses infecting the CNS, the specific mechanisms of herpesvirus infection in the CNS remain poorly understood.

With the continuous development of bioinformatics and high-throughput sequencing technologies, it has become more convenient and efficient to obtain and analyze vast amounts of biological data from complex biological systems [10]. Transcriptome sequencing (RNA-seq) is a genomic research method based on high-throughput sequencing technology, used to comprehensively and systematically analyze the types, quantities, structures, and expression levels of all transcripts (including mRNA, non-coding RNA, etc.) in cells or tissues. In virology research, RNA-seq has been widely applied to study the interactions between viruses and hosts, for example, by revealing the host’s defense strategies against viral infections through differential expression analysis of host transcripts and identifying potential therapeutic targets. Long non-coding RNAs (lncRNAs) account for a large portion of the genome, but their functions are far less well understood than those of mRNAs, possibly due to the cell type-specificity and the complexity of their structures and functions [11]. The advent of RNA-seq has facilitated the study of lncRNAs, and researchers have begun to uncover the critical roles of lncRNAs in the interactions between hosts and viruses [12]. In recent years, numerous studies have shown that during viral infections of host cells or tissues, not only mRNA but also host lncRNA expression is significantly altered. Many lncRNAs have been proven to regulate different steps of the viral infection process and induce host-specific responses to viruses, thereby stimulating antiviral responses or promoting viral replication by inhibiting antiviral reactions [13,14,15,16,17,18]. Although there has been some progress in the field of neurotropic virus infections in recent years, research on lncRNAs related to herpesvirus infections of the CNS, especially neurotropic viruses, is still in its infancy, with limited research findings.

In this study, we used RNA-seq to comprehensively identify mRNAs and long non-coding RNAs (lncRNAs) in the brains of mice infected with PRV. Through a series of bioinformatic analyses, we screened for differentially expressed mRNAs (DE-mRNAs) and differentially expressed lncRNAs (DE lncRNAs). Further, we used Gene Ontology (GO) and Kyoto Encyclopedia of Genes and Genomes (KEGG) enrichment analyses to explore the functions of these differentially expressed genes and constructed a protein–protein interaction network (PPI). The goal of this study is to provide new insights into the pathogenesis of PRV infection in the mouse brain and offer new ideas and directions for research on the pathogenesis of neurotropic virus infections of the CNS.

## 2. Materials and Methods

### 2.1. Mice, Cell Lines, and Virus

Male C57BL/6J mice, 6 weeks old, were purchased from Chengdu Dashuo Experimental Animal Co., Ltd. All mouse experiments were approved by the Animal Ethics Committee of Sichuan Agricultural University (Approval No. 20220261), and all experimental procedures and animal welfare standards strictly followed the guidelines of Sichuan Agricultural University. PK-15 cells (porcine kidney cells) were cultured in Dulbecco’s Modified Eagle Medium (DMEM) (Thermo Fisher Scientific, Waltham, MA, USA) supplemented with 10% fetal bovine serum (FBS, Shenzhen Sunview Technology, Shenzhen, China) and 1% antibiotics (100 U/mL penicillin and 0.1 mg/mL streptomycin; Beyotime Biotechnology, Haimen, China) at 37 °C with 5% CO_2_. The PRV strain SCDJ2024 (GenBank Accession No. PQ189460) used in this study was preserved in our laboratory.

### 2.2. Animal Inoculation

Male C57BL/6J mice, 6 weeks old and weighing approximately 20 g, were housed in an animal room with stable temperature and lighting, with free access to food and water. Mice were randomly divided into two groups: the PRV-inoculated group (*n* = 3) and the mock-inoculated group (*n* = 3). Mice in the PRV-inoculated group were intranasally inoculated with 2.5 μL of virus suspension (2 × 10^5^ PFU) in each nostril, while those in the mock-inoculated group were intranasally inoculated with an equivalent volume of DMEM. Mice in the PRV-inoculated group exhibited severe clinical symptoms, including motor impairment, hunched backs, and intense itching, within 48–72 h post-inoculation (hpi). Mice from both groups were euthanized by CO_2_ asphyxiation at 48, 60, and 72 hpi, and brain samples were collected subsequent for experiments.

### 2.3. DNA Extraction and Viral Copy Number Determination

DNA was extracted from each sample using a DNA extraction kit (Cofitt Biotechnologies, Hong Kong, China) according to the manufacturer’s instructions. The quantity and purity of DNA in each sample were measured using a NanoDrop 2000 spectrophotometer (Thermo Fisher Scientific, Waltham, MA, USA), ensuring an OD260/280 ratio of 1.8. Viral DNA in the samples was quantified by absolute quantification using a standard curve constructed with EP0 DNA, and the viral copy number was calculated.

### 2.4. Total RNA Extraction

Total RNA was extracted from each sample using RNAex Pro RNA extraction reagent (Accurate Biology, Changsha, China) according to the manufacturer’s instructions. The quantity and purity of RNA in each sample were measured using a NanoDrop 2000 spectrophotometer (Thermo Fisher Scientific, Waltham, MA, USA), ensuring an OD260/280 ratio of 2.0.

### 2.5. Library Construction and Sequencing

After qualified RNA samples were obtained, rRNA was removed from the total RNA samples using the TruSeq Stranded Total RNA Library Prep Gold kit (Illumina). RNA sequencing libraries were generated using the NEBNext Ultra Directional RNA Library Prep Kit for Illumina (NEBE7420) according to the manufacturer’s recommendations. Fragmentation buffer was added to the enriched RNA to fragment it into smaller pieces. Using the fragmented RNA as a template, the first strand of cDNA was synthesized with random hexamers, followed by the second strand synthesis with buffer, dNTPs (dTTP in dNTPs was replaced by dUTP), DNA polymerase I, and RNase H. The double-stranded cDNA was purified, end-repaired, A-tailed, and ligated with sequencing adaptors. USER enzyme was used to degrade the second strand of cDNA containing U, followed by PCR enrichment. The final strand-specific libraries were purified using AMPure XP beads. After library validation, different libraries were pooled according to effective concentration and target data volume requirements and sequenced on the NovaSeq X Plus platform.

### 2.6. Quality Control

Raw data in fastq format were processed using an internal Perl script. Clean data were obtained by trimming reads containing adapters, poly-N, or low-quality sequences from the raw data. The Q20, Q30, and GC content of the clean data were calculated. All downstream analyses were based on high-quality clean data.

### 2.7. Read Mapping to the Reference Genome

The reference genome and gene model annotation files were directly downloaded from the genome website. The reference genome index was constructed using HISAT2 v2.0.5 and paired-end clean reads were aligned to the reference genome using HISAT2 v2.0.5 (https://daehwankimlab.github.io/hisat2/, accessed on 1 September 2024).

### 2.8. Gene Expression Level Calculation

Stringtie v1.3.3b (https://ccb.jhu.edu/software/stringtie/, accessed on 1 September 2024) was used to calculate the number of reads mapped to each gene. FPKM (Fragments Per Kilobase of transcript sequence per Million base pairs sequenced) was then calculated.

### 2.9. Differential Expression Analysis

Differential expression analysis was conducted using the edgeR software package v3.22.5 (https://bioconductor.org/packages/release/bioc/html/edgeR.html, accessed on 3 September 2024). The differential expression analysis of mRNA genes and lncRNA transcripts was performed using edgeR. Raw read counts of mRNA genes or lncRNA transcripts across samples were input into the pipeline. The analysis generated normalized read counts, along with normalized mean expression levels between treatment and control groups. A generalized linear model (GLM) was applied to calculate the statistical significance (raw *p*-value) for each gene/transcript. Finally, the Benjamini–Hochberg procedure was employed to adjust for multiple hypothesis testing, yielding the false discovery rate (FDR), typically denoted as padj (adjusted *p*-value). When screening for differentially expressed genes, we adopted the following criteria: |log2FoldChange| ≥ 1 and *p*-values < 0.05.

### 2.10. GO Functional Enrichment and KEGG Pathway Enrichment Analysis

ClusterProfiler (v3.8.1. https://www.bioconductor.org/packages/release/bioc/html/clusterProfiler.html, accessed on 3 September 2024) was used to perform GO functional enrichment and KEGG pathway enrichment analyses on differentially expressed gene sets. The significance level was determined by padj, with a threshold of padj < 0.05 for GO functional enrichment and KEGG pathway enrichment.

### 2.11. lncRNA Target Gene Prediction and Functional Analysis

In this study, lncRNA target genes were predicted based on the co-location or co-expression of lncRNAs and mRNAs. Co-location target gene analysis involved identifying genes within 100 kb upstream and downstream of lncRNAs as potential target genes. Co-expression target gene analysis used a correlation coefficient absolute value greater than 0.95 between lncRNA and mRNA expression as the screening criterion. GO functional enrichment and KEGG pathway enrichment analyses were then performed on the co-location or co-expression target genes of differentially expressed lncRNAs to predict lncRNA functions. A pvalue of less than 0.05 was considered significant for both GO functional enrichment and KEGG pathway enrichment.

### 2.12. Protein–Protein Interaction (PPI) Network Construction and Module Analysis

The differentially expressed genes selected for constructing the PPI network were screened based on the criteria of padj value < 0.05 and |log_2_(FC)| > 1. The STRING database (https://string-db.org, accessed on 15 September 2024) was used to analyze potential interactions among proteins encoded by these differentially expressed genes. The species was set to mouse, and an interaction score > 0.4 was used as the threshold. The PPI network was constructed using Cytoscape software v3.10.0 (https://cytoscape.org/release_notes_3_10_0.html, accessed on 15 September 2024). The MCODE plugin in Cytoscape was used to extract densely connected modules from the PPI network, with a degree cutoff of 2, a node score cutoff of 0.2, a K-score of 2, and a maximum depth of 100.

### 2.13. Quantitative Real-Time PCR (qPCR)

Quantitative real-time PCR (q-PCR) was performed using the CFX Connect™ Real-Time PCR Detection System (Bio-Rad, Hercules, CA, USA). Total RNA was extracted using the AG RNAex Pro RNA extraction reagent (Accurate Biology, Changsha, China) and reverse-transcribed using the Evo M-MLV RT Kit (Accurate Biology, Changsha, China). SYBR Green Pro Taq HS Premix qPCR Kit (Accurate Biology, Changsha, China) was used for qRT-PCR validation. The primers used in this study are listed in Table 1. The reaction conditions were as follows: 95 °C for 2 min, followed by 40 cycles of 95 °C for 5 s, 60 °C for 30 s, 95 °C for 5 s, and 60 °C for 5 s. Relative gene expression was normalized to β-actin expression as an internal standard.

### 2.14. Statistical Analysis

All statistical analyses and graph generation were performed using NovoMagic (https://magic.novogene.com/customer/main#/homeNew, accessed on 1 September 2024) and GraphPad Prism 9.5 (https://www.graphpad.com/updates/prism-950-release-notes, accessed on 17 January 2025). Two-tailed Student’s *t*-tests were used to analyze differences between groups. Data are presented as mean ± SD, with error bars representing at least three independent experiments. For all statistical significance indicators in this manuscript, **** indicates *p* < 0.0001; *** indicates *p* < 0.001; ** indicates *p* < 0.01; * indicates *p* < 0.05; and ns indicates no significance.

## 3. Results

### 3.1. Validation of PRV Infection in Mouse Brains

To prepare samples for RNA-seq analysis, mice were intranasally inoculated with PRV, while the control group was intranasally inoculated with DMEM. Mice in the PRV-inoculated group were euthanized at 48 hpi, 60 hpi, and 72 hpi, and brain samples were collected. The copy number of PRV in mouse brains was detected using quantitative real-time PCR (qRT-PCR). The results showed that after intranasal inoculation with PRV, the virus could rapidly spread to the brain, and the viral load in the brain increased with the extended infection time (Figure 1). We found that the clinical symptoms in mice began to emerge at 48 hpi, and at 60 hpi, the neurological symptoms in mice were most prominent, with the viral load essentially reaching its peak. Ultimately, mouse brains at 60 hpi post-PRV infection were selected for subsequent RNA-seq analysis.

### 3.2. Analysis of Differential Expression Profiles of lncRNAs and mRNAs

We analyzed the differential transcripts between the mock-inoculated group and the PRV-inoculated group using volcano plots and heatmaps (Figure 2). We opted for relatively lenient screening criteria, namely |Log_2_(Fold change)| ≥ 1 and *p* value < 0.05, so as to identify a greater number of differentially expressed genes (DEGs) for analysis and subsequent validation. The results showed significant changes in the expression profiles of lncRNAs and mRNAs in the brain tissues of mice in the PRV-inoculated group compared to the mock-inoculated group. A total of 886 differentially expressed mRNAs and 316 differentially expressed lncRNAs were identified in the PRV-inoculated group compared to the mock-inoculated group, including 549 upregulated and 337 downregulated mRNAs, and 174 upregulated and 142 downregulated lncRNAs.

Subsequently, we performed clustering analysis on the expression levels of the differential transcripts and generated heatmaps (Figure 3). The results showed a significant separation between the PRV infection group and the mock-inoculated group.

### 3.3. GO Enrichment Analysis of Differentially Expressed lncRNAs and mRNAs

We performed GO enrichment analysis in three main ontologies—biological process (BP), cellular component (CC), and molecular function (MF)—to clarify the potential biological functions of differentially expressed transcripts between the PRV infection group and the mock-inoculated group (Figure 4). The results showed that, compared with the mock-inoculated group, the differentially expressed mRNAs in the PRV infection group were enriched in a total of 691 GO terms (padj < 0.05). The results showed that BPs were mainly associated with immune responses. For example, they were mainly enriched in positive regulation of defense response (GO:0031349), response to interferon-beta (GO:0035456), and response to virus (GO:0009615); in the CC ontology, they were enriched in external side of plasma membrane (GO:0009897); and in the MF ontology, they were mainly enriched in growth factor binding (GO:0019838), cytokine activity (GO:0005125), and cytokine binding (GO:0019955). The target genes of differentially expressed lncRNAs predicted by co-expression were enriched in a total of 1173 GO terms (pval < 0.05): in the BP ontology, they were mainly enriched in ncRNA metabolic process (GO:0034660), ncRNA processing (GO:0034470), and RNA-dependent DNA biosynthetic process (GO:0006278); in the CC ontology, they were mainly enriched in nucleolar part (GO:0044452), nuclear transcription factor complex (GO:0044798), and RNA polymerase II transcription factor complex (GO:0090575); and in the MF ontology, they were mainly enriched in RNA-directed DNA polymerase activity (GO:0003964), helicase activity (GO:0004386), and double-stranded RNA binding (GO:0003725). The target genes of differentially expressed lncRNAs predicted by co-location were enriched in a total of 602 GO terms (pval < 0.05): in the BP ontology, they were mainly enriched in response to radiation (GO:0009314), response to light stimulus (GO:0009416), and positive regulation of epithelial cell proliferation (GO:0050679); in the CC ontology, they were mainly enriched in COPI-coated vesicle (GO:0030137), main axon (GO:0044304), and neuromuscular junction (GO:0031594); and in the MF ontology, they were mainly enriched in sodium channel regulator activity (GO:0017080), channel regulator activity (GO:0016247), and transferase activity, transferring glycosyl groups (GO:0016757).

### 3.4. KEGG Enrichment Analysis of Differentially Expressed lncRNAs and mRNAs

We further elucidated the potential biological functions of differentially expressed transcripts between the PRV infection group and the mock-inoculated group through KEGG enrichment analysis (Figure 5). The results showed that, compared with the mock-inoculated group, differentially expressed mRNAs in the PRV infection group were significantly enriched in 35 signaling pathways (padj < 0.05), they were mainly associated with the inflammatory response induced by viral infection, for example, viral protein interaction with cytokine and cytokine receptor (mmu04061), TNF signaling pathway (mmu04668), IL-17 signaling pathway (mmu04657), cytokine–cytokine receptor interaction (mmu04060), and NOD-like receptor signaling pathway (mmu04621). The target genes of differentially expressed lncRNAs predicted by co-expression were significantly enriched in 41 signaling pathways (pval < 0.05), mainly including IL-17 signaling pathway (mmu04657), hepatitis C (mmu05160), cell cycle (mmu04110), ribosome biogenesis in eukaryotes (mmu03008), and nucleocytoplasmic transport (mmu03013). The target genes of differentially expressed lncRNAs predicted by co-location were significantly enriched in 52 signaling pathways (pval < 0.05), mainly including human T-cell leukemia virus 1 infection (mmu05166), human papillomavirus infection (mmu05165), mucin type O-glycan biosynthesis (mmu00512), fluid shear stress and atherosclerosis (mmu05418), and amyotrophic lateral sclerosis (mmu05014).

### 3.5. Construction and Module Analysis of Protein–Protein Interaction (PPI) Network

Differentially Expressed Genes (DEGs) with padj < 0.05 and |log_2_(FC)| > 1 were submitted to the STRING online database to generate the PPI network. This network consisted of 403 nodes and 1993 interaction pairs; we identified key nodes by calculating the degree of nodes. The top 10 high-degree nodes included Tnf, Gapdh, Ccl2, Cxcl10, Icam1, Ccl3, Irf7, Ptgs2, Cd274, and Fcgr3 (Figure 6a). Within this PPI network, we identified 12 modules and selected the top 4 modules with the highest scores for presentation (Figure 6b–e). Functional analysis of DEGs within these 4 modules revealed that these DEGs are primarily involved in viral defense responses, immune responses, and inflammatory responses.

### 3.6. Validation of Differentially Expressed lncRNAs and mRNAs by qRT-PCR

To validate the differential expression of lncRNAs and mRNAs identified by RNA-seq, we selected 42 DEGs (22 mRNAs and 20 lncRNAs) for qRT-PCR analysis (Figure 7). The validated DE-mRNAs include interferon-stimulated genes (*Isg15*, *Usp18*, *Rsad2*, and *Rtp4*), genes related to inflammatory responses (*Tnf* and *Cebpd*), and genes associated with cell death (*Cdkn1a*, *Lcn2*, and *Slc40a*). The qRT-PCR results showed that the expression trends of the selected DEGs were highly consistent with the RNA-seq data, confirming the reliability of the RNA-seq dataset. These findings indicate that PRV infection induces widespread differential expression of lncRNAs and mRNAs in the mouse brain.

## 4. Discussion

Pseudorabies virus (PRV) has a significant impact on the field of veterinary medicine and animal husbandry [19] and may also pose a potential threat to public health [20]. In recent years, researchers have developed a disease animal model that strikingly resembles human herpes simplex encephalitis (HSE) caused by HSV-1 or VZV, using intranasal inoculation of PRV mutants lacking pUL21 and pUS3 [21]. Additionally, due to PRV’s ability to transport from infected neuronal dendrites to axons in a retrograde manner, genetically engineered PRV has been widely used as a live transneuronal tracer for mapping neuronal circuits over the past decade [22]. Although the association between PRV and the nervous system has been studied for decades, research on the molecular mechanisms of PRV infection in the nervous system is still limited. Therefore, to gain a deeper understanding of the central nervous system’s (CNS) response to PRV infection, we identified and analyzed differentially expressed transcripts in PRV-infected mouse brains using RNA-seq technology. A total of 886 mRNAs and 316 lncRNAs were found to be significantly differentially expressed. GO and KEGG enrichment analyses, as well as PPI network construction, revealed that these differentially expressed transcripts are mainly associated with immune and inflammatory responses. In summary, our findings provide a foundation for understanding the biological processes and molecular mechanisms underlying the CNS’s response to PRV infection.

In this study, we observed that at 60 h post-infection (hpi) with PRV, the neurological symptoms in mice were most pronounced, and the viral load had essentially reached its peak. This suggests that at 60 hpi, the impact of PRV on the host nervous system may have reached a climax. Therefore, performing RNA-seq analysis on samples at this time point is more conducive to uncovering the key mechanisms of viral infection. Our analysis of DE-mRNAs from the RNA-Seq results revealed that PRV infection primarily induces innate immune responses in the mouse brain. GO enrichment analysis showed that significantly differentially expressed genes were mainly enriched in the biological processes (BP) of positive regulation of defense response (GO:0031349), cellular response to interferon-beta (GO:0035458), and response to interferon-beta (GO:0035456). These processes mainly include innate immune-related genes, such as interferon-stimulated genes (ISGs). When host cells are infected by viruses, they recognize viral structures and nucleic acids through pattern recognition receptors (PRRs) expressed in the cytoplasm or cell membrane and produce large amounts of interferons (IFNs) via different signaling pathways. IFNs, as the main antiviral cytokines, induce further signaling in cells, leading to various antiviral responses [23,24]. These responses mainly involve the transcriptional upregulation of numerous ISGs with potential antiviral properties [25,26], which are crucial for the host’s defense against viral invasion. Although the CNS has certain immune privilege [27], numerous studies have shown that IFNs play an essential protective role in the CNS against viral infections. Mice lacking IFN-I receptor subunit 1 (Ifnar1−/−) exhibit high susceptibility to multiple viruses, including in the CNS [28,29]. Additionally, some fatal cases of herpes simplex encephalitis (HSE) are associated with defects in key genes in the interferon signaling pathway, such as TRIF, TBK-1, TLR3, and TRAF3 [30,31,32,33]. Major resident cell populations in the CNS, particularly astrocytes and microglia [34,35,36], can sense viral infections and induce the expression of hundreds of ISGs via IFN, generating a certain antiviral response [37]. Our data show significant upregulation of numerous ISGs in the CNS following PRV infection, such as Ifit1, Rsad2, Isg15, Usp18, IFi204, Ch25h, and Ifit2. These significantly upregulated ISGs have been proven to have antiviral effects in many studies. Interferon-stimulated gene 15 (ISG15) is a ubiquitin-like protein induced by IFNα/β. Numerous studies have shown that ISG15 has both pro-viral and antiviral activities, depending on the virus and host species [38,39]. Recently, Liu et al. found that free ISG15 enhances the anti-PRV effects mediated by IFN-α by promoting the phosphorylation and nuclear translocation of STAT1 and STAT2. PRV induces ISG15 expression and ISGylation (ISG15 modification) in the early stages of infection but is subsequently inhibited by the virus, with gE playing a crucial role in reducing ISG15 expression [40]. The IFIT gene family consists of interferon-induced proteins with a tetrapeptide repeat (IFIT). Recent studies have shown that IFIT2 can exert unique antiviral effects in the CNS through various mechanisms. In Ifit2 knockout (Ifit2−/−) mice, VSV activates viral capsid cells in the brain and recruits NK1.1 cells and CD4+ T cells to the brain, facilitating the clearance of neurotropic coronavirus MHV-RSA59 [41]. These findings further confirm the crucial role of IFNs and ISGs in the CNS’s defense against neurotropic viral infections. Additionally, KEGG pathway enrichment analysis revealed that differentially expressed genes were significantly enriched in the Toll-like receptor signaling pathway (mmu04620). Viral infections activate multiple families of pattern recognition receptors (PRRs) in the CNS, including the Toll-like receptor (TLR) family. TLRs recognize pathogen-associated molecular patterns (PAMPs) of viruses and initiate antiviral signaling pathways in the CNS by promoting the transcriptional activation of related genes [42]. Previous studies have shown that TLR3-dependent type I IFN-mediated endogenous antiviral immunity in cortical neurons is crucial for host defense against HSV-1 in the human forebrain [43]. Inherited errors of innate immunity (IEI) caused by TLR3 gene mutations significantly affect type I IFN induction in the CNS [44], weakening its defense against HSV-1. Moreover, transcriptomic analysis of different viral infections in the host brain has shown that differentially expressed genes and enriched pathways are mostly related to innate immune responses and inflammatory responses [45,46,47,48,49]. We also found that the TNF signaling pathway (mmu04668) was significantly enriched in KEGG pathway enrichment analysis. In the protein–protein interaction (PPI) network constructed from differentially expressed genes, the degree centrality of the tumor necrosis factor (TNF) node was the highest. TNF is a central cytokine in inflammatory reactions. It has been found that TNF drives inflammatory responses not only by inducing inflammatory gene expression but also by inducing cell death, instigating inflammatory immune reactions and disease development [50]. Specifically, TNF-α activates downstream pathways such as the MAPK and NF-κB pathways by binding to its receptors TNFR1 and TNFR2. This leads to the transcriptional upregulation of proinflammatory genes, including cytokines (e.g., IL-1β, IL-6) and chemokines (e.g., CXCL8), thereby triggering inflammation. Previous studies have shown that TNF-α is a key mediator of PRV-induced apoptosis. PRV induces apoptosis by activating the p38 MAPK and JNK/SAPK pathways, which increase TNF-α expression and secretion [51]. A recent study demonstrated the crucial role of TNF in viral encephalitis [52]. Using single-cell RNA sequencing (scRNA-seq) and unbiased exploratory gene set enrichment analysis (GSEA), the study showed that HSV-1 infection of human brain organoids activates the TNF-NF-κB signaling pathway, leading to widespread inflammation and brain tissue damage. The combined use of antiviral drug acyclovir (ACV) and anti-inflammatory drugs significantly reduced TNF pathway activation, thereby preventing neuronal damage and maintaining neuroepithelial integrity. Additionally, Xu et al. recently performed transcriptome sequencing on mouse brains following intracranial injection of PRV [53]. The study found significant upregulation of Isg15, Usp18, Oas1b, and Irf7 genes in the PRV infection group. Differentially expressed genes were mainly enriched in innate immune response and inflammatory response in GO terms and in TNF signaling pathway, NOD-like receptor signaling pathway, and IL-17 signaling pathway in KEGG pathways, which is consistent with our findings, indicating that these differentially expressed genes and pathways play important roles in the CNS’s response to PRV infection. However, we also observed some differences, which may arise from variations in PRV infection methods. Compared to intracranial injection, intranasal infection, which we chose in our study, represents a more natural infection route similar to airborne transmission [4]. In this route, PRV first replicates in nasal epithelial cells, then enters the free nerve endings of the trigeminal nerve, replicates in the cell bodies of neurons in the trigeminal ganglia (TG), and can continue to ascend retrogradely from the peripheral nervous system (PNS) to the higher CNS along the nerve axons [54].

Recent studies have shown that the expression levels of some host and viral lncRNAs change during viral infections, and these differentially expressed lncRNAs may promote viral replication or inhibit viral infection by regulating different stages of viral infection or inducing immune responses against viruses [55]. Our study revealed that PRV infection significantly altered the expression of lncRNAs in the host CNS. We first predicted the overall impact of DE-lncRNAs in PRV-infected CNS based on the enrichment analysis of cis-acting and trans-acting target genes of DE-lncRNAs. In the enrichment analysis of cis-acting target genes of lncRNAs, the most significant enrichment was in sodium channel regulator activity (GO:0017080) in GO. Previous studies have shown that latent HSV-1 can change neuronal excitability by regulating the functional activity of sodium channels, thereby participating in the establishment of latency and pain regulation [56]. Therefore, it is reasonable to speculate that these DE-lncRNAs may be associated with herpesvirus latency. In the enrichment analysis of trans-acting target genes of lncRNAs, the most significant enrichments were in the IL-17 signaling pathway (mmu04657), NOD-like receptor signaling pathway (mmu04621), and TNF signaling pathway (mmu04668) in KEGG, which have been proven to be closely related to herpesvirus infections. Previous studies have shown that enhanced Th17 cell differentiation and activation of the IL-17 signaling pathway may be a potential cause of increased risk of herpes zoster in COVID-19 patients [57]. Additionally, recent studies have found that during PRV infection, the expression of the NOD1 gene in BV2 cells significantly increases, and NOD1 participates in cytokine secretion in microglia by regulating the MAPK/NF-κB signaling pathway [58]. It is worth noting that a large part of the results of significantly enriched pathways for trans-acting target genes of DE-lncRNAs and DE-mRNAs are the same, which well illustrates that DE-lncRNAs can exert functions by regulating DE-mRNAs. In fact, the functions of lncRNAs predicted by the aforementioned co-expression and co-localization methods have certain limitations. First, co-expression relationships do not necessarily imply direct functional associations, as there may be indirect regulation or spurious correlations. Second, co-expression analysis relies on high-quality transcriptomic data, and noise and batch effects in the data may affect the accuracy of the results. Furthermore, although subcellular localization is closely related to the function of lncRNAs, studying the subcellular localization of lncRNAs within cells solely through co-localization analysis still makes it difficult to determine their precise mechanisms of action. In addition, some lncRNAs may have different localizations under different cellular states or conditions, which increases the complexity of functional prediction. Most importantly, predicting the functions of lncRNAs solely through bioinformatic analysis is incomplete, as lncRNAs, in addition to regulating gene expression in cis or trans, can also participate in post-transcriptional regulation, mRNA processing, splicing, transport, and translation [59]. Subsequent integrative analysis of multi-omics data (for example, the translatome, proteome, and metabolome) can enhance the accuracy of lncRNA function prediction and provide certain directions for lncRNA function validation. In recent years, the identification of lncRNAs associated with PRV infection has also made some progress. During the process of PRV infection of PK15 cells, LNCA02830 is significantly upregulated. LNCA02830 can promote PRV replication by inhibiting the innate immune response through significantly reducing the mRNA levels of IRF3, IFNβ, and MX1 [60]. In addition, a recent study has shown that LNC641 in 3D4/21 cells is significantly upregulated after PRV infection. It has been verified that LNC641 can downregulate the production of IFN-α by inhibiting the JAK/STAT1 pathway, thereby suppressing the innate immune response to PRV infection and increasing PRV replication [13]. In our data, we also found several DE-lncRNAs that have been reported to be associated with viral infections, such as NEAT1. A recent study observed significant overexpression of NEAT1 in the blood of COVID-19 patients and found that NEAT1 is significantly correlated with cytokines such as IL-6, CCL2, and TNF-α. Finally, based on multivariate regression analysis, NEAT1 was identified as an independent predictor of overall survival (*p* = 0.02) [61]. NEAT1 can also regulate innate immune responses induced by viral infections. For example, after HTNV infection, NEAT1 transcription is activated via the RIG-I-IRF7 pathway. NEAT1 eliminates the transcriptional repression of splicing factor proline and glutamine-rich protein (SFPQ) by relocalizing SFPQ to paraspeckles, thereby promoting the expression of RIG-I and DDX60, enhancing IFN-β production, and inhibiting HTNV infection [62]. NEAT1 is also a key nuclear regulator of innate immune response activation mediated by the cGAS-STING pathway. NEAT1 can interact with HEXIM1 to form a HEXIM1-DNA-PK-paraspeckle component ribonucleoprotein complex (HDP-RNP), which in turn interacts with cGAS and its partner PQBP1. Their interactions are reshaped by exogenous DNA, ultimately leading to the release of paraspeckle proteins, recruitment of STING, and activation of DNAPKc and IRF3 [63]. Additionally, HSV-1 infection increases NEAT1 expression and perinuclear speckle formation in a STAT3-dependent manner. NEAT1 and other perinuclear speckle protein components, such as P54nrb and PSPC1, can bind to HSV-1 genomic DNA. PSPC1 recruits STAT3 to perinuclear speckles by binding to STAT3 and promotes the interaction between STAT3 and viral gene promoters, ultimately increasing viral gene expression and viral replication [64]. In recent years, numerous studies have demonstrated that lncRNAs can influence viral replication by participating in innate immune responses or through other mechanisms. The study found that lncBST2-2 is significantly upregulated by various viral infections through the IFN/JAK/STAT signaling pathway and promotes innate immune responses to viral infections by targeting IRF3 [65]. LncBST2-2 inhibits the replication of multiple viruses (such as HCV, NDV, VSV, and HSV) and enhances IFN production by promoting the phosphorylation, dimerization, and nuclear translocation of IRF3. LncBST2-2 also interacts with the DNA-binding domain of IRF3, enhancing the binding of TBK1 to IRF3, thereby inducing robust IFN production. Moreover, lncBST2-2 blocks the interaction between IRF3 and the PP2A-RACK1 complex, which is essential for the dephosphorylation of IRF3. Other studies have found that the primate-specific lncRNA *CHROMR* is induced by infections of influenza A virus and SARS-CoV-2 and coordinates the expression of interferon-stimulated genes (ISGs) that execute the antiviral response. Expression of *CHROMR* is crucial for restricting viral infections in macrophages, and *CHROMR* can license IRF-dependent signaling and transcription of the ISGs network by sequestering the interferon regulatory factor (IRF)-2-dependent transcriptional corepressor IRF2BP2 [14]. Recently, lncRNA-BTX was found to be upregulated upon infection with various viruses [15]. LncRNA-BTX promotes the demethylation of DHX9, enhancing its interaction with JMJD6, thereby facilitating the translocation of DHX9 from the nucleus to the cytoplasm, where it can bind to viral RNA and promote viral replication. Concurrently, lncRNA-BTX inhibits the exposure of the NES domain of ILF3 by enhancing the interaction between ILF3 and ILF2, preventing its translocation from the nucleus to the cytoplasm and reducing its inhibitory effect on viral replication. Additionally, lncRNAs have also been recently discovered to directly inhibit viral replication. Researchers used RNA-Seq to identify a subset of lncRNAs with putative antiviral activity against CHIKV, including the previously uncharacterized antiviral lncRNA prohibiting human alphaviruses (*ALPHA*). Although *ALPHA* is induced by multiple alphavirus infections, it specifically inhibits a subset of alphaviruses, namely CHIKV and ONNV. This is because *ALPHA* acts independently of the canonical IFN response and directly binds to the CHIKV genomic RNA to inhibit viral RNA replication [18]. There has also been some progress in understanding the role of lncRNAs in the CNS’s response to viral infections. The lncRNA EDAL is induced by various neurotropic viruses infecting host neurons. EDAL can positively regulate the transcription of Pcp4l1, which encodes a 10 kDa peptide that inhibits the replication of multiple neurotropic viruses. Therefore, EDAL can indirectly inhibit the replication of these neurotropic viruses in neuronal cells and rabies virus infection in mouse brains [17]. Additionally, another study found that linc-AhRA, through its conserved 117-nucleotide fragment, acts as a molecular scaffold to bind tripartite motif-containing protein 27 (TRIM27) and TANK-binding kinase 1 (TBK1), thereby enhancing the interaction between TRIM27 and TBK1. This interaction promotes TRIM27-mediated ubiquitination of TBK1 and leads to the degradation of TBK1 via the ubiquitin–proteasome pathway, ultimately restricting the innate immune response of microglia to neurotropic herpesvirus infections. Recent studies have also shown that lncRNA JINR1, produced during JEV infection, can regulate the transcriptional activity of NF-κB via RBM10 or directly interact with the p65 subunit of NF-κB, thereby playing an important role in promoting flavivirus replication and flavivirus-induced neuronal cell death [66].

In summary, our findings provide new insights into the functional characterization of DE-mRNAs and DE-lncRNAs involved in the CNS’s response to PRV infection. However, further mechanistic studies are needed to fully identify and understand these potentially biologically significant DE-mRNAs and DE-lncRNAs. Therefore, based on this study, future experiments will explore the targeting regulatory mechanisms between lncRNAs and mRNAs and the interaction mechanisms between different neural cells and viruses.

## 5. Conclusions

In this study, we identified 886 differentially expressed mRNAs (DE-mRNAs) and 316 differentially expressed lncRNAs (DE-lncRNAs) through RNA-seq. GO and KEGG enrichment analyses and PPI network analysis revealed that these differentially expressed transcripts are primarily involved in immune responses. Our research is the first to report the expression profile of lncRNAs in the CNS following PRV infection. However, further in-depth studies are required to elucidate the roles of these lncRNAs during viral infection. In summary, this study provides a foundation for a comprehensive understanding of the molecular mechanisms underlying the CNS’s response to PRV infection and offers direction for future research.

## Figures and Tables

**Figure 1 viruses-17-00580-f001:**
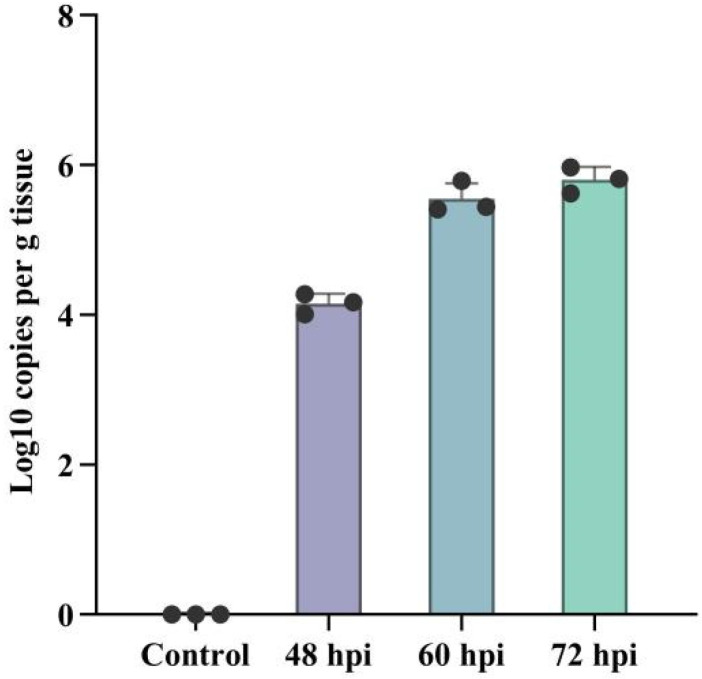
Validation of PRV infection in mouse brains by qPCR. The copy number of the PRV EP0 gene in mouse brains at 48, 60, and 72 hpi was quantified by quantitative real-time PCR (q-PCR). Error bars represent the standard deviation between replicates. Data are presented as mean ± SD based on three independent experiments.

**Figure 2 viruses-17-00580-f002:**
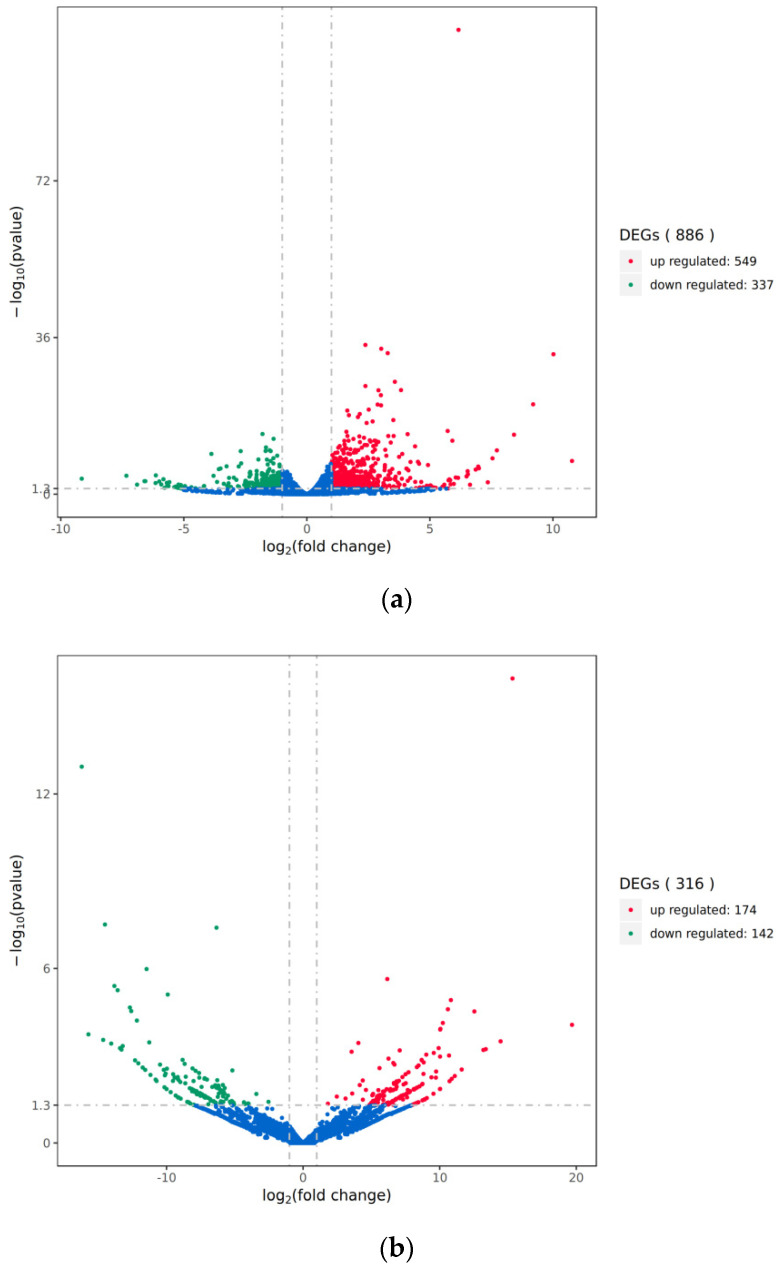
Volcano plot of differential transcripts between PRV infection group and mock-inoculated group. (**a**) Volcano plot of differentially expressed mRNAs between the PRV infection group and the mock-inoculated group. (**b**) Volcano plot of differentially expressed lncRNAs between the PRV infection group and the mock-inoculated group. In each plot, the *x*-axis represents log_2_(fold change), and the *y*-axis represents -log_10_(pvalue). Each dot represents a transcript, with red dots indicating upregulated transcripts and green dots indicating downregulated transcripts. The total number of upregulated and downregulated transcripts is labeled on the right side of each plot.

**Figure 3 viruses-17-00580-f003:**
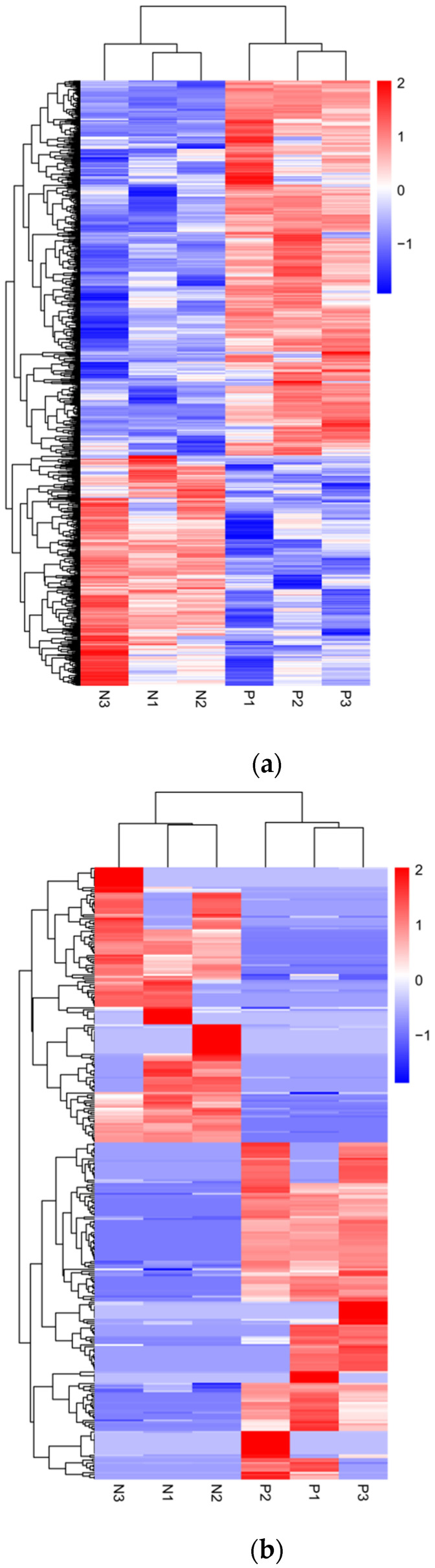
Hierarchical clustering analysis of differential transcript expression levels between PRV infection group and mock-inoculated group. (**a**) Heatmap of differentially expressed mRNA levels between the PRV infection group and the mock-inoculated group. (**b**) Heatmap of differentially expressed lncRNA levels between the PRV infection group and the mock-inoculated group. In each heatmap, the color intensity represents the levels of differential transcript expression, with color gradients from blue to red indicating downregulated to upregulated expression levels.

**Figure 4 viruses-17-00580-f004:**
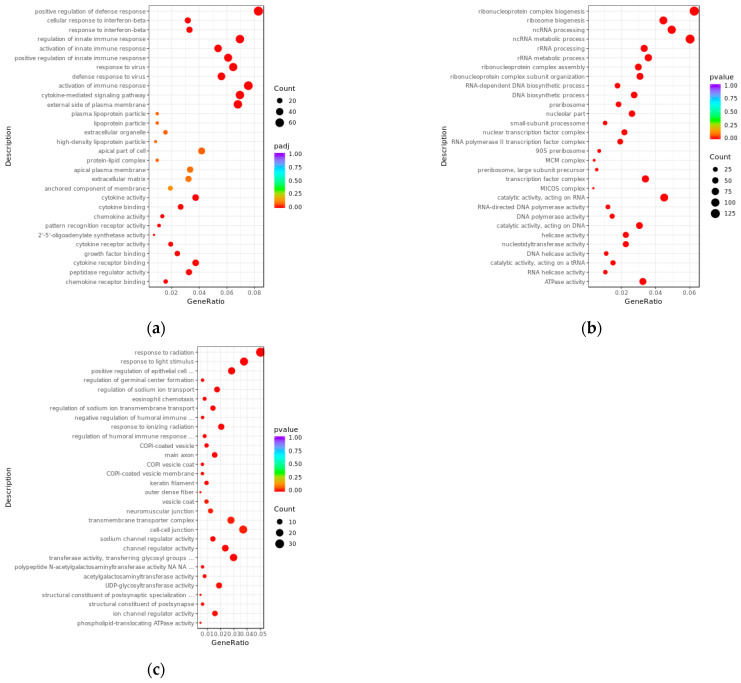
Bubble plots of GO enrichment analysis for differentially expressed transcripts between PRV infection group and mock-inoculated group. (**a**) Bubble plot of GO enrichment analysis for differentially expressed mRNAs, with padj < 0.05. (**b**) Bubble plot of GO enrichment analysis for target genes of differentially expressed lncRNAs predicted by co-expression, with pvalue < 0.05. (**c**) Bubble plot of GO enrichment analysis for target genes of differentially expressed lncRNAs predicted by co-location, with pvalue < 0.05. In the plots, the *x*-axis (GeneRatio) represents the ratio of the number of target genes enriched in a specific pathway to the total number of target genes. The *y*-axis indicates the biological processes or pathways where the functions are exerted. The size of the bubbles reflects the number of transcripts enriched in GO terms, and the color of the bubbles corresponds to different ranges of padj or pval.

**Figure 5 viruses-17-00580-f005:**
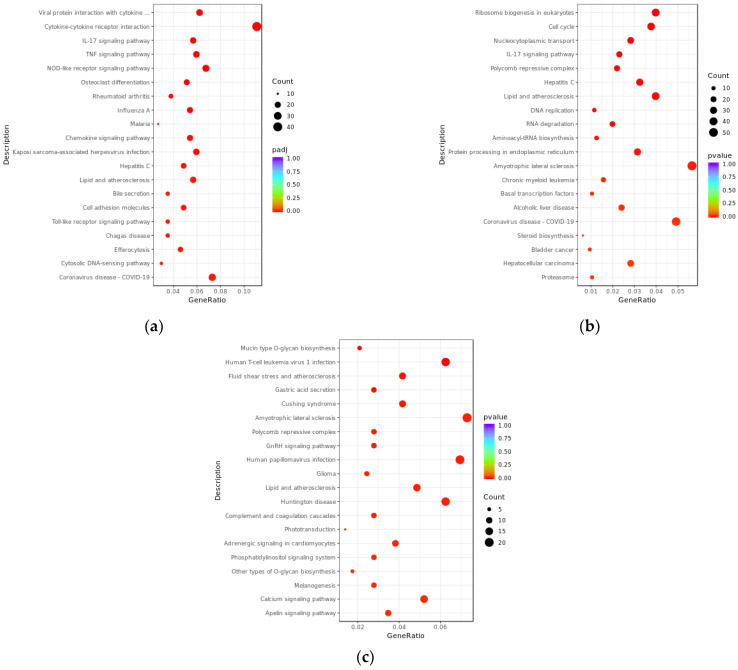
KEGG enrichment analysis bubble plots of differentially expressed transcripts between PRV infection group and mock-inoculated group. (**a**) Bubble plot of KEGG enrichment analysis for differentially expressed mRNAs, with padj < 0.05. (**b**) Bubble plot of KEGG enrichment analysis for target genes of differentially expressed lncRNAs predicted by co-expression, with pvalue < 0.05. (**c**) Bubble plot of KEGG enrichment analysis for target genes of differentially expressed lncRNAs predicted by co-location, with pvalue < 0.05. In the plots, the *x*-axis (GeneRatio) represents the ratio of the number of target genes enriched in a specific pathway to the total number of target genes. The *y*-axis indicates the specific pathways that are enriched. The size of the bubbles reflects the number of transcripts enriched in KEGG pathways, and the color of the bubbles corresponds to different ranges of padj or pval.

**Figure 6 viruses-17-00580-f006:**
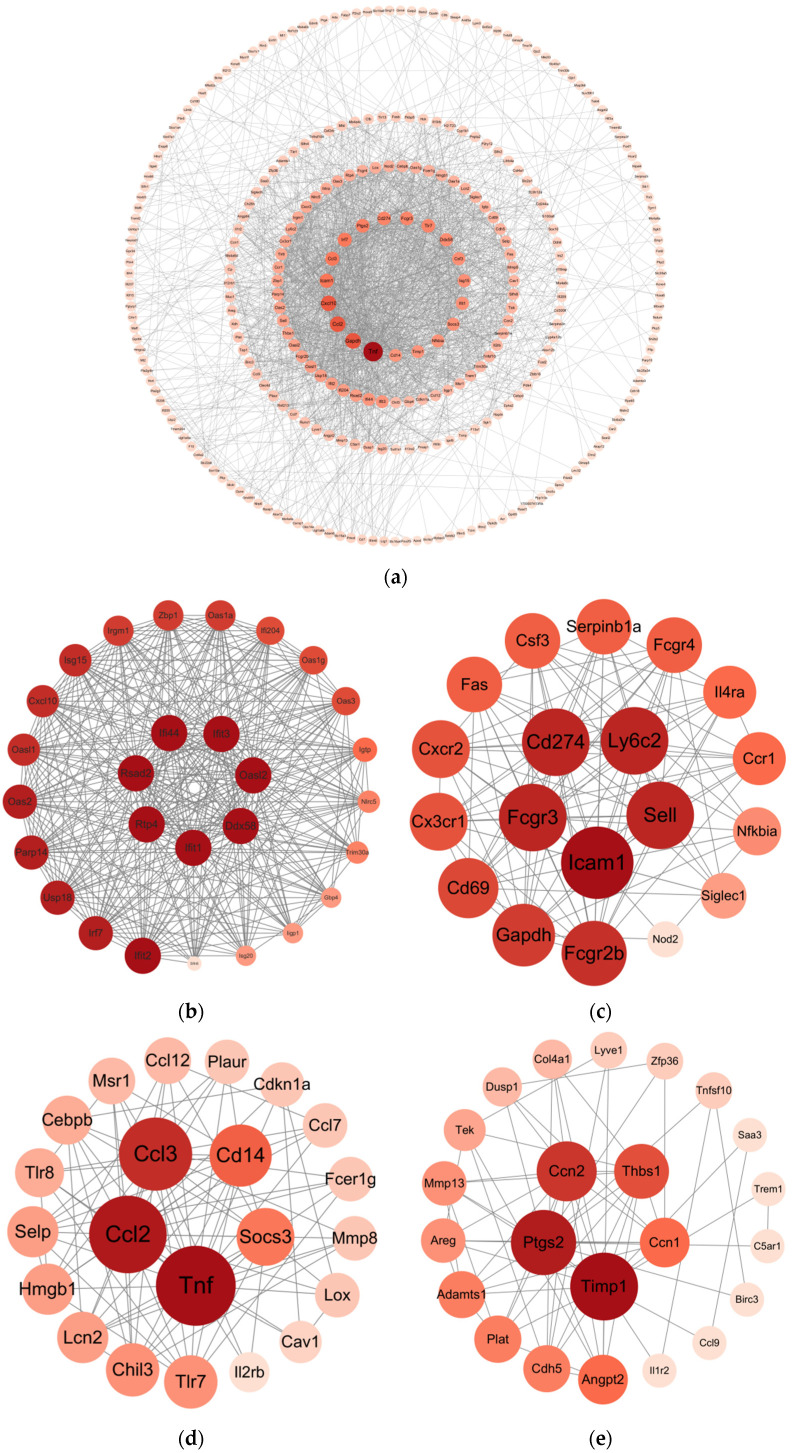
Construction and module extraction of PPI network. (**a**) The overall PPI network generated from DEGs, comprises 403 nodes and 1993 edges, with an average node degree of 9.89, an average local clustering coefficient of 0.442, an expected number of edges of 557, and a PPI enrichment *p*-value < 1.0 × 10^−16^. (**b**) Module 1, composed of 28 nodes and 335 interaction pairs. (**c**) Module 2, composed of 19 nodes and 115 interaction pairs. (**d**) Module 3, composed of 22 nodes and 94 interaction pairs. (**e**) Module 4, composed of 23 nodes and 67 interaction pairs. The MCODE plugin in Cytoscape was used to extract densely connected modules from the PPI network, with the following parameters: degree cutoff = 2, node score cutoff = 0.2, K-score = 2, and maximum depth = 100. Each node represents DEGs, and each edge indicates the interaction between the proteins encoded by two genes.

**Figure 7 viruses-17-00580-f007:**
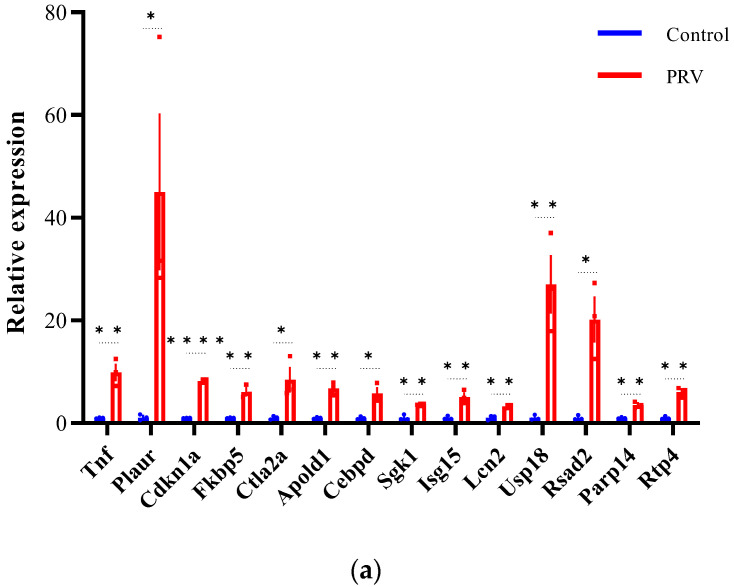
Validation of differentially expressed mRNAs and lncRNAs by qRT-PCR (**a**–**d**). Expression of mRNA transcripts and lncRNA transcripts was measured by qRT-PCR and data are represented as means ± S.D. and analyzed using unpaired two-tailed *t*-test. Statistical significance is indicated as follows: ****, *p* < 0.0001; **, *p* < 0.01; *, *p* < 0.05; and ns, no significance.

**Table 1 viruses-17-00580-t001:** Primers for q-PCR.

**Primers**	**Sequences (5′-3′)**
LINC375-F	5′ CTGCCCTAATGTTATCCC 3′
LINC375-R	5′ GCCATCTTACAGCCACC 3′
Msl2-IT2-F	5′ AGTGACTGCCATCTTCT 3′
Msl2-IT2-R	5′ CACAACCCTGTAACCTA 3′
LINC49-F	5′ GTTCTGTCGCTTGGTAAT 3′
LINC49-R	5′ CTGGAGTTGGAGTGGG 3′
Mest-OT9-F	5′ ATGAATCTACCTCGTGC 3′
Mest-OT9-R	5′ CTCTTCTCCTAACTCCCT 3′
Msl2-IT1-F	5′ AGTGACTGCCATCTTCT 3′
Msl2-IT1-R	5′ CACAACCCTGTAACCTA 3′
LINC402-F	5′ GAACCCTCCCAACTAAA 3′
LINC402-R	5′ TCTGCCAAAGGAATCTA 3′
LINC240-F	5′ CCTCACCCTAAAGTTCG 3′
LINC240-R	5′ CTGGGCAGGGAAAGTA 3′
LINC377-F	5′ CTGCCCTAATGTTATCCC 3′
LINC377-R	5′ GCCATCTTACAGCCACC 3′
LINC355-F	5′ TGGTTCTGTCGCTTGG 3′
LINC355-R	5′ CTGGAGTTGGAGTGGG 3′
LINC156-F	5′ GCCTCCTGGAACAACC 3′
LINC156-R	5′ AACTGGGCAGGGAAAG 3′
NR_166531.1-F	5′ CCCAAACCAAATCCAC 3′
NR_166531.1-R	5′ ACAATCACCTGGCACAT 3′
LINC351-F	5′ TTCCCAGAACACTCCTAA 3′
LINC351-R	5′ AGTCTCGACCGTCAGC 3′
LINC155-F	5′ GCCTCCTGGAACAACC 3′
LINC155-R	5′ AACTGGGCAGGGAAAG 3′
XR_004939660.1-F	5′ GTTACGGAAAGCATCTC 3′
XR_004939660.1-R	5′ CAGAGCAAGTGGGTCA 3′
XR_387385.5-F	5′ ATCCGCCTTCTTGTGC 3′
XR_387385.5-R	5′ TGTTTGATGGTGGTGTCG 3′
Ccdc171-IT1-F	5′ GAACCCTCCCAACTAAA 3′
Ccdc171-IT1-R	5′ TCTGCCAAAGGAATCTA 3′
XR_387382.1-F	5′ ATCCGCCTTCTTGTGC 3′
XR_387382.1-R	5′ TGTTTGATGGTGGTGTCG 3′
XR_381404.4-F	5′ GGATGTGATGTGGAGGTT 3′
XR_381404.4-R	5′ GCATTCAGTGGCTTCTATT 3′
LINC192-F	5′ CCTCACCCTAAAGTTCAA 3′
LINC192-R	5′ AAGGAGTACCTGGTTCAT 3′
XR_001783316.3-F	5′ ATCTGTATCAGTTAGGGTTC 3′
XR_001783316.3-R	5′ TTGCCAGCACATCTTT 3′
Plaur-F	5′ CGGGAATGGCAAGATGA 3′
Plaur-R	5′ TCTGGTCCAAAGAGGTGC 3′
Cdkn1a-F	5′ TTCCGCACAGGAGCAAA 3′
Cdkn1a-R	5′ AAGTCAAAGTTCCACCGTTCT 3′
Fkbp5-F	5′ ATGAGGGCACCAGTAACAA 3′
Fkbp5-R	5′ CCAAGGCTAAAGGCAAAT 3′
Ctla2a-F	5′ TGAGCAGGGCAAGACC 3′
Ctla2a-R	5′ TCAGGCAAATCAGGAG 3′
Apold1-F	5′ GTGCGGAGGGTGCAGGAGAT 3′
Apold1-R	5′ AGCCGAAGAAGACGATGAAGTAGA 3′
Cebpd-F	5′ CCTGCCATGTACGACGACGAG 3′
Cebpd-R	5′ GCCGCTTTGTGGTTGCTGTT 3′
Sgk1-F	5′ GGGCACATCGTCCTCACT 3′
Sgk1-R	5′ CGGTCATACGGCTGCTTA 3′
Isg15-F	5′ CGATTTCCTGGTGTCCGTGA 3′
Isg15-R	5′ CTCGCTGCAGTTCTGTACCA 3′
Usp18-F	5′ CGTGCCGTTGTTTGTC 3′
Usp18-R	5′ GGCTTTGCGTCCTTATC 3′
Rsad2-F	5′ CCCTCTGTGAGCATAGTGA 3′
Rsad2-R	5′ GCCACCTTGTAATCCCT 3′
Parp14-F	5′ AGATTGGTCGTCAGTTCG 3′
Parp14-R	5′ GCTATGTCCCTCTGTAGGT 3′
Rtp4-F	5′ CCCCGATGACTTCAGTAC 3′
Rtp4-R	5′ CCTGAGCAGAGGTCCAAC 3′
Tnf-qF	5′ CTCATTCCTGCTTGTGGC 3′
Tnf-qR	5′ CACTTGGTGGTTTGCTACG 3′
P2ry12-qF	5′ GCTTGGCAATGAGGAT 3′
P2ry12-qR	5′ GGTAGCGGTCAATGGTT 3′
Slco1a4-qF	5′ GGTTGCCTGCTGCTCT 3′
Slco1a4-qR	5′ TTCCGTTCTCCATCATTCT 3′
Itm2a-qF	5′ GCCCAAGAGCACCATT 3′
Itm2a-qR	5′ TTCCCTTCTCAAAGTCGT 3′
Serpinb1a-qF	5′ ATACCCTCAACTCTAACCTG 3′
Serpinb1a-qR	5′ AATGTAGCAATGCCTCC 3′
Pltp-qF	5′ TCTGCCCTGTGCTCTACC 3′
Pltp-qR	5′ GGCTCCAGTTGTCCTCCTT 3′
Pllp-qF	5′ CGAAAGTGAGCACGAGGAC 3′
Pllp-qR	5′ CACCAGCCAGAGGAAGACA 3′
Slc40a1-qF	5′ TTGGCTTGCTGGTATTG 3′
Slc40a1-qR	5′ AACAGATGTATTCGGTTGAT 3′
Gpr34-qF	5′ CGTGGGACTGGTTGGA 3′
Gpr34-qR	5′ GGGAGGCAGAAGATGA 3′
Lcn2-F	5′ GCCCAGGACTCAACTCA 3′
Lcn2-R	5′ GCCCACAACGTACCACC 3′
Actb-F	5′ AATCGTGCGTGACATCAA 3′
Actb-R	5′ AGAAGGAAGGCTGGAAAA 3′
PRV-EP0-F	5′ CGGGCGAAGACAAACAAAGG 3′
PRV-EP0-R	5′ GGGCGGTAGAAGCCAAAGATC 3′

## Data Availability

All available data are presented in the article.

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
