# Peer review of "Expression Profiles of lncRNAs and mRNAs in the Mouse Brain Infected with Pseudorabies Virus: A Bioinformatic Analysis"

_viruses, 2025, doi:10.3390/v17040580_

Round 1
Reviewer 1 Report
Comments and Suggestions for Authors
The manuscript titled "Expression Profiles of lncRNAs and mRNAs in Mouse Brain Infected with Pseudorabies Virus and Bioinformatics Analysis" is an original article providing first insights of the differential expression of long non-coding RNAs (lncRNAs) and mRNAs in the brains of mice infected with Pseudorabies virus (PRV). The study employs RNA sequencing (RNA-seq) and bioinformatics tools to identify and analyze the roles of these transcripts in PRV infection. The study is innovative. But we have some minor comments:
- The manuscript does not provide details on the statistical methods used for differential expression analysis. Clarifying these methods would enhance the rigor of the study.
- While the manuscript discusses the potential roles of lncRNAs in viral infections, it does not delve deeply into the specific mechanisms by which these lncRNAs might influence PRV infection or host immunity.
- The discussion could be expanded to include more detailed hypotheses about how the identified lncRNAs might regulate immune responses or viral replication.
Author Response
Reviewer 1
The manuscript titled "Expression Profiles of lncRNAs and mRNAs in Mouse Brain Infected with Pseudorabies Virus and Bioinformatics Analysis" is an original article providing first insights of the differential expression of long non-coding RNAs (lncRNAs) and mRNAs in the brains of mice infected with Pseudorabies virus (PRV). The study employs RNA sequencing (RNA-seq) and bioinformatics tools to identify and analyze the roles of these transcripts in PRV infection. The study is innovative. But we have some minor comments:
- The manuscript does not provide details on the statistical methods used for differential expression analysis. Clarifying these methods would enhance the rigor of the study.
- While the manuscript discusses the potential roles of lncRNAs in viral infections, it does not delve deeply into the specific mechanisms by which these lncRNAs might influence PRV infection or host immunity.
- The discussion could be expanded to include more detailed hypotheses about how the identified lncRNAs might regulate immune responses or viral replication.
Comment 1:
The manuscript does not provide details on the statistical methods used for differential expression analysis. Clarifying these methods would enhance the rigor of the study.
Reponse 1:
Thank you for your important suggestions regarding the methodological rigor. In response, we have supplemented the revised manuscript with detailed information on the statistical methods used for differential expression analysis, with the specific modifications located in lines 155-165.
Thank you once again for your contribution to enhancing the quality of this manuscript!
Comment 2:
While the manuscript discusses the potential roles of lncRNAs in viral infections, it does not delve deeply into the specific mechanisms by which these lncRNAs might influence PRV infection or host immunity.
Reponse 2:
Thank you for your meticulous review of this manuscript and for the valuable comments you have provided. The questions you raised regarding the specific mechanisms by which long non-coding RNAs (lncRNAs) are involved in pseudorabies virus (PRV) infection are of great importance. We fully agree with your perspective and have supplemented and improved the relevant content in the revised manuscript, with the specific modifications located in lines 552-560.
Since the majority of differentially expressed lncRNAs (DE-lncRNAs) in our data are functionally uncharacterized, further experimental validation is required to explore their specific mechanisms of action in PRV infection or host immunity. In the discussion section, we have added information on lncRNAs that have been experimentally validated to be associated with PRV.
Thank you once again for your contribution to enhancing the quality of this manuscript!
Comment 3:
The discussion could be expanded to include more detailed hypotheses about how the identified lncRNAs might regulate immune responses or viral replication.
Reponse 3:
Thank you for your meticulous review of this manuscript and for the valuable comments you have provided. The questions you raised regarding the specific mechanisms by which long non-coding RNAs (lncRNAs) are involved in pseudorabies virus (PRV) infection are of great importance. We fully agree with your perspective and have supplemented and improved the relevant content in the revised manuscript, with the specific modifications located in lines 552-560.
Since the majority of differentially expressed lncRNAs (DE-lncRNAs) in our data are functionally uncharacterized, further experimental validation is required to explore their specific mechanisms of action in PRV infection or host immunity. In the discussion section, we have added information on lncRNAs that have been experimentally validated to be associated with PRV.
Thank you once again for your contribution to enhancing the quality of this manuscript!

Reviewer 2 Report
Comments and Suggestions for Authors
The manuscript validated the differential expression trends of lncRNAs and mRNAs using quantitative real-time PCR, the findings provide new insights into the roles of lncRNAs and mRNAs during PRV infection of the host CNS. There are some problems that should be addressed.
(1) It is suggested that the sample size of mice infection experiment be further discussed. In this paper, each group is only n=3, is it enough to reflect biological variation? Is there sample size estimation or repeated experiments to enhance the credibility of the conclusion?
(2) Three time points, 48 hours, 60 hours and 72 hours, were selected in the article, but in the end, 60 hours of data were selected by RNA‑seq. It is suggested that the author explain why 60 hours is chosen as the key time point, and discuss the possible dynamic trend of transcriptome changes at different time points.
(3) edgeR was used in differential expression analysis, but only the threshold of p‑value was given. It is suggested to explain whether multiple tests (such as FDR or padj) were carried out to reduce false positives, and the basis for setting the threshold was clear.
(4) The analysis results of GO and KEGG are rich, but the descriptions of some items are lengthy. It is suggested that the author further integrate and highlight the key pathways, and discuss the correlation between these pathways and PRV infection and immune response of nervous system.
PPI Network Construction and Module Analysis
(5) For the construction of PPI network and module extraction, it is suggested to supplement the basis for selecting more parameters and the biological significance of key nodes inside the module, especially the specific mechanism of highly connected nodes (such as TNF).
(6) Co-expression and co-location methods are used to predict the function of lncRNA, but these two methods have their own limitations. It is suggested to objectively evaluate the reliability of the prediction in the discussion part, and put forward the direction of subsequent experimental verification.
(7) The results of qRT-PCR verification are consistent with RNA‑seq data, but only 15 DEG verification are listed in this paper. It is suggested to increase the verification of more key lncRNA or mRNA, or to discuss why these 15 representative genes were selected.
(8) The resolution of some heat maps and volcano maps in the map is low, so it is suggested to improve the quality of the chart, and explain the meanings of colors and marks in detail in the legend, so that readers can intuitively understand the data distribution and statistical significance.
Comments on the Quality of English Languagegood
Author Response
Reviewer 2
The manuscript validated the differential expression trends of lncRNAs and mRNAs using quantitative real-time PCR, the findings provide new insights into the roles of lncRNAs and mRNAs during PRV infection of the host CNS. There are some problems that should be addressed.
- It is suggested that the sample size of mice infection experiment be further discussed. In this paper, each group is only n=3, is it enough to reflect biological variation? Is there sample size estimation or repeated experiments to enhance the credibility of the conclusion?
- Three time points, 48 hours, 60 hours and 72 hours, were selected in the article, but in the end, 60 hours of data were selected by RNAseq. It is suggested that the author explain why 60 hours is chosen as the key time point, and discuss the possible dynamic trend of transcriptome changes at different time points.
- edgeR was used in differential expression analysis, but only the threshold of pvalue was given. It is suggested to explain whether multiple tests (such as FDR or padj) were carried out to reduce false positives, and the basis for setting the threshold was clear.
- The analysis results of GO and KEGG are rich, but the descriptions of some items are lengthy. It is suggested that the author further integrate and highlight the key pathways, and discuss the correlation between these pathways and PRV infection and immune response of nervous system.
- PPI Network Construction and Module Analysis. For the construction of PPI network and module extraction, it is suggested to supplement the basis for selecting more parameters and the biological significance of key nodes inside the module, especially the specific mechanism of highly connected nodes (such as TNF).
- Co-expression and co-location methods are used to predict the function of lncRNA, but these two methods have their own limitations. It is suggested to objectively evaluate the reliability of the prediction in the discussion part, and put forward the direction of subsequent experimental verification.
- The results of qRT-PCR verification are consistent with RNAseq data, but only 15 DEG verification are listed in this paper. It is suggested to increase the verification of more key lncRNA or mRNA, or to discuss why these 15 representative genes were selected.
- The resolution of some heat maps and volcano maps in the map is low, so it is suggested to improve the quality of the chart, and explain the meanings of colors and marks in detail in the legend, so that readers can intuitively understand the data distribution and statistical significance.
Comment 1:
It is suggested that the sample size of mice infection experiment be further discussed. In this paper, each group is only n=3, is it enough to reflect biological variation? Is there sample size estimation or repeated experiments to enhance the credibility of the conclusion?
Reponse 1:
Thank you for your meticulous review of the experimental design of this manuscript and your professional suggestions regarding the sample size. The issues you raised about biological variability and statistical power are of great importance. In this study, the mouse infection experiments were conducted with a biological replicate size of n=3 per group, based on the following considerations:
Firstly, in previous models of acute herpesvirus infection, the infection phenotypes (such as viral load and expression of inflammatory factors) typically show high consistency among mice with the same genetic background (C57BL/6J) (Laval K, Vernejoul JB, Van Cleemput J, Koyuncu OO, Enquist LW. Virulent Pseudorabies Virus Infection Induces a Specific and Lethal Systemic Inflammatory Response in Mice. J Virol. 2018 Nov 27;92(24):e01614-18. doi: 10.1128/JVI.01614-18. PMID: 30258005; PMCID: PMC6258956.).
Secondly, a biological replicate size of 3 has been adopted in numerous similar studies (Zhao P, Zhao L, Zhang K, Feng H, Wang H, Wang T, Xu T, Feng N, Wang C, Gao Y, Huang G, Qin C, Yang S, Xia X. Infection with street strain rabies virus induces modulation of the microRNA profile of the mouse brain. Virol J. 2012 Aug 11;9:159. doi: 10.1186/1743-422X-9-159. PMID: 22882874; PMCID: PMC3549733.; Sui B, Chen D, Liu W, Tian B, Lv L, Pei J, Wu Q, Zhou M, Fu ZF, Zhang Y, Zhao L. Comparison of lncRNA and mRNA expression in mouse brains infected by a wild-type and a lab-attenuated Rabies lyssavirus. J Gen Virol. 2021 Mar;102(3). doi: 10.1099/jgv.0.001538. Epub 2020 Dec 7. PMID: 33284098.).
Moreover, in our preliminary experiments, to rule out randomness, we repeated the infection experiments in different batches of C57 mice and observed stable changes in viral load and the expression levels of key host antiviral-related genes (such as ISGs) at 60 hours post-infection (hpi), indicating minimal inter-individual variability. This supports the current sample size (n=3) in capturing significant changes and further validates the reliability of the subsequent sequencing results. We acknowledge that a larger sample size (e.g., n=5−6) would yield more accurate data, but a sample size of n=3 is sufficient for the initial screening of data with potential biological significance in this study.
Thank you once again for your valuable suggestions to enhance the quality of this manuscript!
Comment 2:
Three time points, 48 hours, 60 hours and 72 hours, were selected in the article, but in the end, 60 hours of data were selected by RNAseq. It is suggested that the author explain why 60 hours is chosen as the key time point, and discuss the possible dynamic trend of transcriptome changes at different time points.
Reponse 2:
We sincerely appreciate your in-depth review of the experimental design of this manuscript and the valuable comments you have provided. Your questions regarding the rationality of the time point selection and the dynamic trends of the transcriptome are of great significance.
In response to your queries, we have supplemented the discussion section in the revised manuscript, with the specific modifications located in lines 422-427 of the manuscript. Our choice of 60 hours post-infection (hpi) as the key time point is primarily based on the following considerations:
Firstly, our preliminary experiments revealed that the expression levels of host antiviral genes peak at 60 hpi. Secondly, clinical symptoms in mice begin to appear at 48 hpi, and by 60 hpi, the neurological symptoms in mice are most pronounced, with viral load reaching near-maximal levels. Lastly, the aim of our study is to initially screen for mRNAs, lncRNAs, and potential biological processes involved in the host central nervous system's response to PRV infection, providing a basis for further functional validation. Based on these points, we selected 60 hpi as the critical time point for data collection, which aligns with our research objectives.
Once again, thank you for your contribution to enhancing the quality of this manuscript!
Comment 3:
edgeR was used in differential expression analysis, but only the threshold of pvalue was given. It is suggested to explain whether multiple tests (such as FDR or padj) were carried out to reduce false positives, and the basis for setting the threshold was clear.
Reponse 3:
Thank you for your rigorous review of the differential expression analysis methods in this manuscript and for your valuable suggestions. The questions you raised regarding multiple testing correction and threshold setting are of vital importance.
In response to your comments, we have made additional explanations in the revised manuscript, specifically in lines 233-236. In this study, we used uncorrected p-values (p < 0.05) as the initial screening threshold for differential expression analysis, based on the following considerations:
Firstly, the aim of this study is to broadly identify candidate differentially expressed mRNAs (DE-mRNAs) and lncRNAs (DE-lncRNAs) and construct regulatory networks to provide references for further experimental validation. When we applied a strict false discovery rate (FDR) correction (e.g., FDR < 0.05), we found that the number of differentially expressed genes was too small, which would lead to insufficient power for functional enrichment analysis.
Additionally, we set an absolute fold change threshold (|log2FC| > 1) to exclude differences with low biological significance (such as minor expression fluctuations), thereby trying to strike a balance between false negatives and false positives. We acknowledge that not using FDR correction may increase the risk of false positives. However, the design goal of this study is to conduct an exploratory screening of these differentially expressed genes (DEGs). Therefore, we chose a relatively lenient p-value as the threshold to identify more candidate DEGs for subsequent analysis and experimental validation. We have carefully weighed the pros and cons of false positives and false negatives.
We recognize that this approach may introduce some false positives, but we believe that the potential for discovering new and meaningful biological insights outweighs this risk. Moreover, we will rigorously validate the selected genes through experiments in the follow-up studies to confirm their significance.
Thank you once again for your contribution to improving the quality of this manuscript!
Comment 4:
The analysis results of GO and KEGG are rich, but the descriptions of some items are lengthy. It is suggested that the author further integrate and highlight the key pathways, and discuss the correlation between these pathways and PRV infection and immune response of nervous system.
Reponse 4:
We sincerely appreciate your meticulous review of the functional enrichment analysis section and your valuable suggestions. Your proposal to integrate key pathways and further explore their associations with PRV infection and neuroimmunity is highly constructive.
In response to your suggestions, we have made additional modifications in the discussion section of the revised manuscript, specifically in lines 277-280, 318-325, and 428-497. We have integrated key pathways when describing the analysis results. In the discussion section, we have provided a detailed exploration of the mechanisms by which these key pathways are involved in the PRV infection process and their interrelationships with immune response of nervous system.
However, regarding the more extensive results from the enrichment analysis that were not discussed in the text, we believe they still hold some biological significance. Therefore, we have chosen to retain them in the manuscript rather than delete them. We hope you understand our rationale.
Thank you once again for your contribution to enhancing the logical coherence and scientific depth of this manuscript!
Comment 5:
PPI Network Construction and Module Analysis. For the construction of PPI network and module extraction, it is suggested to supplement the basis for selecting more parameters and the biological significance of key nodes inside the module, especially the specific mechanism of highly connected nodes (such as TNF).
Reponse 5:
Thank you for your in-depth review of the PPI network analysis and for your valuable suggestions. Your questions regarding the basis for parameter selection and the biological mechanisms of key nodes are of great importance. In response, we have supplemented the revised manuscript with additional information, specifically in lines 351-353, 366-369, and 477-497. We have provided more details about the parameters used in constructing the PPI network and extracting modules. In the discussion section, we have also elaborated on the biological significance and mechanisms of action of the key node TNF.
Thank you once again for your outstanding contribution to enhancing the scientific depth of this manuscript!
Comment 6:
Co-expression and co-location methods are used to predict the function of lncRNA, but these two methods have their own limitations. It is suggested to objectively evaluate the reliability of the prediction in the discussion part, and put forward the direction of subsequent experimental verification.
Reponse 6:
Thank you for your important corrections regarding the methodological limitations of this manuscript. In response, we have added new content to the discussion section of the revised manuscript, specifically in lines 536-552. Here, we have detailed the shortcomings of using co-expression and co-localization to predict lncRNA functions and have proposed directions for subsequent validation.
Thank you once again for your contribution to improving the quality of this manuscript!
Comment 7:
The results of qRT-PCR verification are consistent with RNAseq data, but only 15 DEG verification are listed in this paper. It is suggested to increase the verification of more key lncRNA or mRNA, or to discuss why these 15 representative genes were selected.
Reponse 7:
Thank you for your valuable suggestions regarding the rigor of experimental validation. In response to the issues you raised, we have supplemented the experimental data in the results section of the revised manuscript, specifically in lines 376-401. We have added validation for the downregulated differentially expressed genes to further demonstrate the reliability of the omics data, including eight downregulated lncRNAs and eight downregulated mRNAs. Additionally, we have provided some explanations for the validated DE-mRNAs. In the discussion section, we have elaborated on the roles of some validated differentially expressed genes in the viral infection process.
Thank you once again for your contribution to improving the quality of this manuscript!
Comment 8:
The resolution of some heat maps and volcano maps in the map is low, so it is suggested to improve the quality of the chart, and explain the meanings of colors and marks in detail in the legend, so that readers can intuitively understand the data distribution and statistical significance.
Reponse 8:
Thank you for your valuable suggestions regarding the quality of the figures in our study. In response to this issue, we have re-uploaded higher-resolution images, with the specific modifications reflected in lines 243-247 and 259-262. Additionally, we have provided detailed explanations for the information presented in the figures.
Thank you once again for your contribution to improving the quality of this manuscript!

Round 2
Reviewer 2 Report
Comments and Suggestions for Authors
I have no other comments